# Clinical characteristics of the "Gap" between the prevalence and incidence of hearing loss using National Health Insurance Service data

**Junhun Lee**[1,2], **Chul Young Yoon** [1,2], **Juhyung Lee**[1,2], **Tae Hoon Kong**[1,2,3], **Young Joon Seo** [1,3]*

**1** Research Institute of Hearing Enhancement, Yonsei University Wonju College of Medicine, Wonju, South Korea, **2** Department of Medical Informatics and Biostatistics, Yonsei University Wonju College of Medicine, Wonju, South Korea, **3** Department of Otorhinolaryngology, Yonsei University Wonju College of Medicine, Wonju, South Korea

* okas2000@hanmail.net

**Data Availability Statement:** For reasons such as personal information protection, registration of health insurance data is restricted due to domestic laws in Korea, and we conducted the study by

## Abstract

### Objectives

Hearing loss is the inability to hear speech or sounds well, owing to a number of causes. This study aimed to simultaneously determine the prevalence, incidence, and the Gap between them in hearing loss in South Korean patients at the same point in time as well as to identify patients who have not recovered from hearing loss.

### Methods

We examined the prevalence and incidence of patients diagnosed with hearing loss in the National Health Insurance Service database over an 11-year period from 2010 to 2020. The difference between the prevalence and the incidence was defined in this study as the term "Gap". Gap is the number of patients converted into the number of patients per 100,000 people by subtracting the incidence from the prevalence. Clinical characteristics such as sex and age per 100,000 individuals were examined.

### Results

As of 2020, the domestic prevalence obtained in this study was 1.84%, increasing annually, and the prevalence increased with age to 4.10% among those over 60. The domestic incidence was 1.57%, increasing annually, and the incidence increased with age to 3.36% for those over 60s. The Gap was 0.27%, showing a steady increase from 2011 to 2020 with a corresponding increase in insurance benefit expenses.

### Conclusion

To fully understand the burden of hearing loss and develop effective prevention and treatment strategies, it is important to measure the Gap between its prevalence and incidence. This Gap means a lot because hearing loss is an irreversible disease. Gap represents patients who have already been diagnosed with hearing loss and are being diagnosed every

obtaining permission to use health insurance data. The data analyzed in this study was obtained from the Korean National Health Insurance Service (NHIS). The following licenses/restrictions apply: only Korean researchers can access these datasets. Requests to access these datasets should be directed to NHIS, https://nhiss.nhis.or.kr/bd/ab/bdaba000eng.do.

**Funding:** The present study was grant-funded by three institutions supported by the Korean government. - The National Research Foundation of Korea (No. NRF-2020R1A2C1009789) - The Korean Fund for Regenerative Medicine (21C0721L1) - The Commercialization Promotion Agency for R&D Outcomes (2023, 1711199152) The funders had no role in study design, data collection and analysis, decision to publish, or preparation of the manuscript.

**Competing interests:** The authors have declared that no competing interests exist.

year, indicating that the number of patients who do not recover is increasing. In other words, the increase in Gap meant that there were many patients who constantly visited the hospital for diagnosis of hearing loss.

## Introduction

Hearing loss is the inability to hear speech or sounds well, owing to a number of causes. It can occur for a variety of reasons, including genetic factors, age-related changes, exposure to loud noises, infections, and certain medications. Therefore, hearing loss can be categorized as conductive, sensorineural, or mixed. Conductive hearing loss is caused by mechanical problems in sound transmission from the environment to the inner ear through the eardrum and ossicles. Sensorineural hearing loss is the most common type of hearing loss in adults in primary care settings and can result from cochlear or retrocochlear changes [1]. Understanding the incidence and prevalence of the different types of hearing loss is important for several reasons: identifying at-risk populations for healthcare providers to better target their screening and prevention efforts; to enable researchers and healthcare providers to develop more effective treatments and interventions; and for public health officials to plan and implement initiatives to improve hearing health in the general population [2].

The terms prevalence and incidence are used to describe different aspects of the occurrence of a health condition, such as hearing loss. Prevalence refers to the total number or proportion of individuals in a population with a certain condition either at a specific point in time or over a certain period. Incidence, on the other hand, refers to the number of new cases of a health condition that occur within a specified period of time. Both prevalence and incidence are important measures for understanding the impact of hearing loss and developing strategies to prevent, diagnose, and treat the condition [3].

According to the most recent estimates, 500 million people worldwide suffer from hearing impairment. In terms of monetary value, it costs the world more than $750 billion annually [4]. In the United States, an estimated 23% of Americans over the age of 12 are affected by hearing loss. Older adults have a higher prevalence of hearing loss and report more severe levels of hearing loss [5]. In fact, a study of 5,742 U.S. adults aged 20–69 years from 1999 to 2004 found that 16.1% experienced hearing loss [6]. Germany has a prevalence of 16–25% in systematic literature searches [7]. A cohort study in France estimated the prevalence among 18- to 75-year-olds to be 25% [8]. In India, meta-analyses reported a prevalence of hearing loss ranging from 6 to 26.9% [9]. Using data from the Korean National Health and Nutrition Examination Survey (KNHANES) from 2009 to 2012, unilateral hearing loss, defined as a unilateral hearing loss of less than 41 dB in a hearing-impaired ear with a pure tone average of at least 41 dB assessed at 0.5, 1.0, 2.0, and 3.0 kHz, was reported to be 5.55% [10]. A study using the same data estimated that the prevalence of unilateral and bilateral hearing loss in adolescents aged 13–18 years was 2.2% and 0.4%, respectively [11]. Studies using National Health Insurance data, have analyzed the prevalence of moderate-to-severe hearing loss using the National Disability Register. Consequently, it decreased from 0.5% in 2006 to 0.46% in 2015 [12]. However, there is a lack of research using large-scale data to determine the prevalence and incidence of hearing loss using operational definitions.

The Gap between the prevalence and incidence refers to the difference between the number of people with health conditions at a given point in time (prevalence) and the number of new cases diagnosed within a specific period (incidence). Some people may not realize that they

have hearing loss or may choose not to seek treatment, which can lead to a Gap between the prevalence and incidence rates. Another reason for the Gap between the prevalence and incidence is that some people with the condition may recover or improve over time, while new cases continue to be diagnosed. This can result in a higher prevalence rate than incidence rate. For example, some types of hearing loss, such as conductive hearing loss caused by ear infection, may improve over time with treatment, leading to a lower incidence rate than prevalence rate.

We sought ways to utilize data without test records to identify patients who did not recover from hearing loss. We found no prior studies comparing prevalence and incidence using big healthcare data. Prevalence and incidence can be used to determine the prevalence of a disease and gather information regarding prevention and treatment options. Therefore, this study aimed to simultaneously determine the prevalence, incidence, and the Gap between them in hearing loss in South Korean patients at the same point in time as well as to identify patients who have not recovered from hearing loss.

## Materials and methods

The study was conducted from 2022-07-01 to 2023-11-30, and the study period was from 2021-07-06 to 2023-12-31. Data is available from 2021-12-14 to 2023-12-13.

### Study population and data collection

This study was approved by the Institutional Review Board (IRB) of Yonsei University Wonju Severance Christian Hospital (CR321338) for human subject research.

This study was conducted using national data provided by the National Health Insurance Service (KNHIS). The Korean National Health Insurance Service (KNHIS) data includes the entire population of South Korea, as 97% of South Koreans have health insurance. These data are billed as treatment, with the major drawback that they are not linked to each hospital's electronic medical record (EMR), leaving no test records [13]. It collects medical history, drug prescription, and medical examination information of health insurance-eligible people (nationwide), de-identifies the information subjects so that they cannot be recognized, and subsequently provides datasets for policy and academic research. Health and medical academics (associations, universities, research institutes, etc.) can utilize National Health Information Data to conduct systematic research activities related to national health.

For this study, it was customized in the NHIS database and included as a criterion for patients diagnosed with hearing loss in the main or sub-diagnosis from 2010 to 2020. Each patient must also have personal information such as birth year and gender. Excluding non-reimbursable treatment for 11 years in Korea, the total number of patients extracted by diagnosis of hearing loss was 6,424,491.

The studies were conducted in accordance with the local legislation and institutional requirements. The IRB waived the requirement of written informed consent for participation from the participants or the participants' legal guardians/next of kin because this study is a study using data collected retrospectively and the data collected is not data collected for research.

### Experimental design

Data were extracted based on the type of hearing loss, using the presence or absence of an International Classification of Diseases 10th Revision (ICD-10) diagnosis code (main/sub). The following ICD10 codes were used: conductive hearing loss (H90.0, H90.1, and H90.2), sensorineural hearing loss (H90. 3, H90.4, H90.5), mixed hearing loss (H90.6, H90.7, H90.8),

ototoxicity (H91.0), presbycusis hearing loss (H91.1), sudden hearing loss (H91.2), noise-induced hearing loss (H83.3), and other hearing loss (H91.9) [14]. The diagnosis name according to the classified type of hearing loss was included in the study subjects if they were treated at least once from 2010 to 2020. Therefore, there are patients who have received multiple types of hearing loss diagnoses per patient included in the study.

This study was not included in the analysis in NHIS customized data when it was claimed afterwards for reasons such as aviation unions, occupational soldiers, overseas stay, VIP (national agency), and long-term leave. In addition, a small number of patients were excluded due to errors in notation such as gender and age. The number of study subjects extracted after excluding 640,062 from a total of 6,424,491 customized data was 5,784,429. This study investigated and compared the prevalence and incidence of patients defined as hearing loss over 11 years from 2010 to 2020, and defined the difference as "Gap" in this study. Finally, we analyzed the total number of patients in Gap and the number of patients by type (Fig 1).

In order to obtain Gap, it is necessary to clarify the operational definition of prevalence and incidence. The numerator of prevalence included patients diagnosed with hearing loss in the year. The numerator of incidence was included in the corresponding year of the date of re-diagnosis of hearing loss after 365 days of the interval between diagnostic records after the first diagnosis or diagnosis. Both denominators were calculated as the total population of the Republic of Korea for the year. The prevalence minus the incidence rate was calculated as Gap and converted into the number of patients per 100,000 people. Clinical characteristics such as sex and age per 100,000 individuals were examined. Age groups were categorized as follows: < 10, 10s, 20s, 30s, 40s, 50s, and ≥ 60s. The average annual growth rate (CAGR) was calculated by averaging the growth rate by year during the analysis period.

### Participant selection

Statistical analysis was performed using SAS software version 9.4 (SAS Institute, Cary, NC).

Multiple regression analysis was used to determine which type of patient affected the total number of Gap patients. We excluded ototoxicity, presbycusis, and noise-induced hearing loss with a small number of patients. Additionally, we used the ARIMA model, a time series analysis model, to determine whether seasonality exists in the number of patients by type of Gap. We divided each year into four seasons: March, April, and May into Spring; June, July, and August into Summer; September, October, and November into Fall; and December, January, and February into Winter since 2010. Two-sided analysis was performed, and a P value of less than 0.05 was considered to indicate significance.

## Results

### The current state of hearing loss prevalence

The number of patients with hearing loss has steadily increased over the past 11 years from 604,702 in 2010 to 942,764 in 2020. The prevalence of hearing loss per 100,000 people increased from 1,212.3 in 2010 to 1,836.0 in 2020, with a compound annual growth rate of 4.24%. Of these, male patients increased from 1,103.6 in 2010 to 1,668.0 in 2020 per 100,000, a 4.22% increase, and female patients increased from 1,321.4 in 2010 to 2,003.1, a 4.25% compound annual growth rate. By age group, the average annual increase in prevalence over the 11-year period was -2.33%, 2.59%, 5.40%, 5.12%, 3.12%, 1.15%, and 1.87% for those under 10, 10s, 20s, 30s, 40s, 50s, and over 60s, respectively (Table 1).

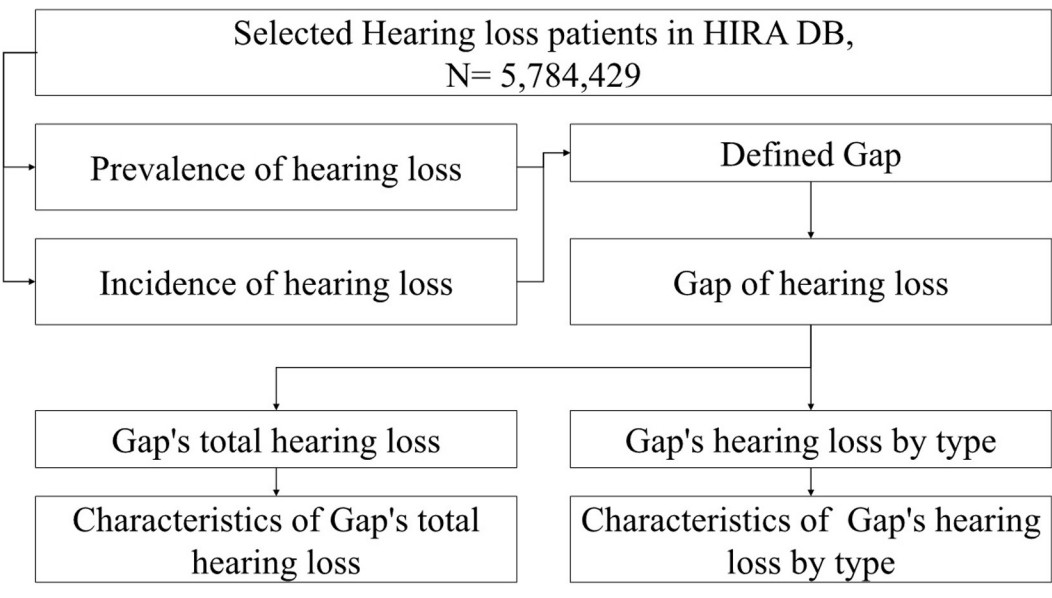

**Fig 1.**

### The current state of hearing loss incidence

The number of people with hearing loss has increased steadily over the past decade, from 556,879 in 2011 to 806,397 in 2020, and the incidence of hearing loss per 100,000 people has increased from 1,111.3 in 2010 to 1,570.4 in 2020, a compound annual growth rate of 3.92%. Of these, the number of male patients increased by 4.06%, from 994.7 in 2011 to 1,423.2 in 2020 per 100,000 people, and the number of female patients increased by 3.92%, from 1,214.4 in 2011 to 1,716.8 in 2020. By age group, the average annual increase in incidence over 10 years was -4.22%, 1.33%, 4.43%, 4.96%, 3.08%, 1.18%, and 1.76% for those under 10, 10s, 20s, 30s, 40s, 50s, and over 60s, respectively (Table 2).

**Table 1. Prevalence of patients with hearing loss, 2010–2020 (per 100,000).**

| Year | 2011 | 2012 | 2013 | 2014 | 2015 | 2016 | 2017 | 2018 | 2019 | 2020 |
|---|---|---|---|---|---|---|---|---|---|---|
| Number of patients | 604,702 | 626,844 | 638,193 | 659,202 | 671,299 | 732,067 | 810,000 | 844,955 | 892,085 | 973,078 |
| Per 100,000 | 1,212.3 | 1,250.9 | 1,267.6 | 1,303.8 | 1,322.4 | 1,436.8 | 1,584.7 | 1,649.3 | 1,738.9 | 1,895.5 |
| Age (%) | | | | | | | | | | |
| <10 | 528.1 | 552.1 | 543.5 | 541.8 | 515.5 | 493.2 | 504.6 | 487.0 | 481.8 | 552.3 |
| 10–19 | 542.8 | 616.5 | 611.1 | 618.3 | 621.8 | 638.1 | 692.5 | 725.4 | 689.0 | 721.0 |
| 20–29 | 612.3 | 694.3 | 715.9 | 746.9 | 769.1 | 813.9 | 879.5 | 935.1 | 931.5 | 969.1 |
| 30–39 | 691.1 | 730.9 | 771.4 | 790.3 | 815.7 | 867.7 | 937.6 | 990.7 | 1,017.6 | 1,105.4 |
| 40–49 | 909.9 | 929.4 | 934.7 | 950.5 | 962.6 | 1,019.7 | 1,083.4 | 1,113.0 | 1,126.3 | 1,220.9 |
| 50–59 | 1,580.2 | 1,563.9 | 1,540.2 | 1,528.4 | 1,520.8 | 1,619.7 | 1,686.3 | 1,696.7 | 1,743.9 | 1,837.9 |
| ≥60 | 3,405.2 | 3,325.4 | 3,271.0 | 3,314.7 | 3,271.8 | 3,560.1 | 3,951.2 | 4,005.0 | 4,231.0 | 4,511.5 |
| Gender (%) | | | | | | | | | | |
| Male | 1,103.6 | 1,120.1 | 1,139.9 | 1,172.2 | 1,183.1 | 1,291.8 | 1,431.6 | 1,487.6 | 1,567.6 | 1,715.1 |
| Female | 1,321.4 | 1,366.5 | 1,395.5 | 1,435.5 | 1,461.7 | 1,581.6 | 1,737.5 | 1,810.6 | 1,909.6 | 2,075.0 |

**Table 2. Incidence of patients with hearing loss, 2011–2020 (per 100,000).**

| Year | 2011 | 2012 | 2013 | 2014 | 2015 | 2016 | 2017 | 2018 | 2019 | 2020 |
|---|---|---|---|---|---|---|---|---|---|---|
| Number of patients | 556,879 | 565,793 | 583,684 | 593,276 | 651,389 | 714,564 | 740,802 | 779,756 | 845,413 | 806,397 |
| Per 100,000 | 1,111.3 | 1,123.8 | 1,154.5 | 1,168.7 | 1,278.4 | 1,398.0 | 1,446.0 | 1,520.0 | 1,646.8 | 1,570.4 |
| Age (%) | | | | | | | | | | |
| <10 | 470.1 | 459.0 | 451.0 | 425.9 | 406.6 | 418.2 | 398.9 | 394.4 | 453.9 | 318.9 |
| 10–19 | 553.5 | 544.6 | 546.8 | 552.2 | 572.2 | 623.1 | 649.9 | 612.5 | 641.3 | 623.4 |
| 20–29 | 649.2 | 666.4 | 692.2 | 714.9 | 758.7 | 818.6 | 868.7 | 861.5 | 895.2 | 958.9 |
| 30–39 | 679.2 | 716.8 | 732.7 | 756.3 | 808.9 | 873.2 | 922.6 | 943.9 | 1,022.7 | 1,050.2 |
| 40–49 | 848.1 | 851.8 | 864.8 | 876.6 | 930.9 | 989.2 | 1,014.0 | 1,023.1 | 1,106.3 | 1,114.7 |
| 50–59 | 1,395.4 | 1,373.5 | 1,361.4 | 1,354.5 | 1,453.0 | 1,499.2 | 1,503.4 | 1,547.8 | 1,622.3 | 1,550.8 |
| ≥60 | 2,871.2 | 2,819.4 | 2,864.6 | 2,812.1 | 3,093.0 | 3,386.6 | 3,400.0 | 3,588.9 | 3,798.9 | 3,359.8 |
| Gender (%) | | | | | | | | | | |
| Male | 994.7 | 1,010.5 | 1,037.8 | 1,044.6 | 1,150.2 | 1,262.3 | 1,303.6 | 1,368.4 | 1,488.5 | 1,423.2 |
| Female | 1,214.4 | 1,237.3 | 1,271.1 | 1,292.8 | 1,406.5 | 1,533.4 | 1,588.0 | 1,671.0 | 1,804.3 | 1,716.8 |

## Gap between the prevalence and incidence of hearing loss

When we plotted the prevalence and incidence by year, we observed an increasing trend for both prevalence and incidence (Fig 2). We defined the Gap (Prevalence-Incidence rates) and found that the number of people in the Gap increased steadily over the decade, from 69,965 in 2011 to 136,367 in 2020. The Gap per 100,000 people with hearing loss increased from 139.6 in 2011 to 265.6 in 2020, with a compound annual growth rate of 7.41% (Table 3).

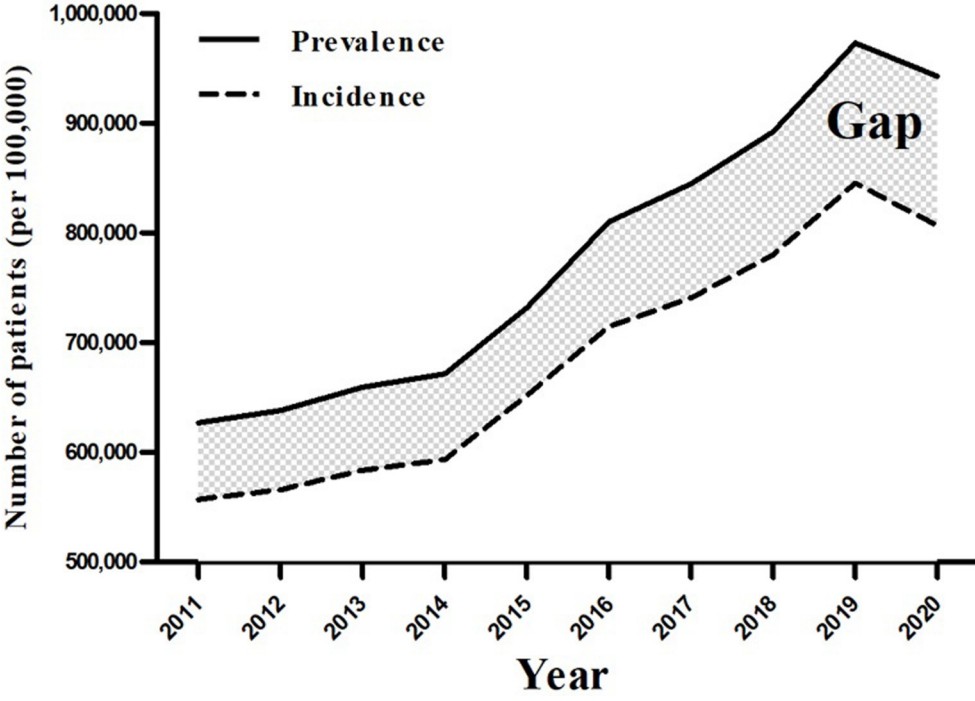

**Fig 2.**

Table 3. Gaps between prevalence and ncidence of hearing loss, 2011–2020 (per 100,000).

| Year | 2011 | 2012 | 2013 | 2014 | 2015 | 2016 | 2017 | 2018 | 2019 | 2020 |
|---|---|---|---|---|---|---|---|---|---|---|
| Number of patients | 69,965 | 72,400 | 75,518 | 78,023 | 80,678 | 95,436 | 104,153 | 112,329 | 127,665 | 136,367 |
| Per 100,000 | 139.6 | 143.8 | 149.4 | 153.7 | 158.3 | 186.7 | 203.3 | 219.0 | 248.7 | 265.6 |
| Age (%) | | | | | | | | | | |
| <10 | 82.1 | 84.4 | 90.9 | 89.6 | 86.6 | 86.3 | 88.1 | 87.4 | 98.5 | 98.3 |
| 10–19 | 63.0 | 66.5 | 71.5 | 69.6 | 65.9 | 69.4 | 75.5 | 76.6 | 79.8 | 77.8 |
| 20–29 | 45.1 | 49.5 | 54.7 | 54.2 | 55.3 | 60.9 | 66.5 | 70.0 | 73.8 | 77.5 |
| 30–39 | 51.6 | 54.6 | 57.6 | 59.4 | 58.8 | 64.5 | 68.1 | 73.7 | 82.7 | 88.3 |
| 40–49 | 81.3 | 83.0 | 85.7 | 86.0 | 88.8 | 94.1 | 99.0 | 103.2 | 114.6 | 122.3 |
| 50–59 | 168.5 | 166.7 | 167.0 | 166.3 | 166.6 | 187.1 | 193.2 | 196.2 | 215.6 | 220.9 |
| ≥60 | 454.2 | 451.6 | 450.1 | 459.8 | 467.1 | 564.6 | 605.0 | 642.1 | 712.7 | 739.4 |
| Gender (%) | | | | | | | | | | |
| Male | 125.5 | 129.4 | 134.4 | 138.5 | 141.6 | 169.3 | 183.9 | 199.2 | 226.5 | 244.7 |
| Female | 152.1 | 158.2 | 164.4 | 168.9 | 175.1 | 204.1 | 222.6 | 238.6 | 270.7 | 286.3 |

## The current state of Gap

The annual Gap was identified based on sex and age. The highest number of patients were in their 60s and older, but the age group with the highest average annual growth rate was individuals in their 20s. The average annual increase was in the following order:20s (6.20%), 30s (6.14%), 60s (5.56%), 40s (4.64%), 50s (3.06%), 10s (2.36%), and under 10s (2.02%). We looked at the results of calculating the number of patients per 100,000 by age and gender. Age-specific Gap rates were higher for those over 60s and under 10 years of age than for patients in their 10s, 20s, and 30s. Among men, the rate increased from 125.5 per 100,000 in 2011 to 244.7 per 100,000 in 2020, a 7.71% increase, and among women, the rate increased from 152.1 per 100,000 in 2011 to 286.3 per 100,000 in 2020, a 7.28% increase (Table 3).

The annual Gap was identified based on sex and age. The highest number of patients were in their 60s and older, but the age group with the highest average annual growth rate was individuals in their 20s. The average annual increase was in the following order:20s (6.20%), 30s (6.14%), 60s (5.56%), 40s (4.64%), 50s (3.06%), 10s (2.36%), and under 10s (2.02%). We looked at the results of calculating the number of patients per 100,000 by age and gender. Age-specific Gap rates were higher for those over 60s and under 10 years of age than for patients in their 10s, 20s, and 30s. Among men, the rate increased from 125.5 per 100,000 in 2011 to 244.7 per 100,000 in 2020, a 7.71% increase, and among women, the rate increased from 152.1 per 100,000 in 2011 to 286.3 per 100,000 in 2020, a 7.28% increase (Table 3).

To understand the meaning of the Gap between prevalence and incidence, we calculated Gap and the annual the total medical expenses. Similar to the Gap's growth rate, the total medical expenses for Gap patients increased from KRW 4.5 billion in 2011 to KRW 14.8 billion in 2020, a compound annual growth rate of 13.96%. Until 2015, the Gap remained constant. However, in 2016, it began to widen significantly. In fact, from 2012 to 2015, the growth rate was 2–3% per year, followed by 17.92% in 2016, 8.88% in 2017, 7.77% in 2018, 13.57% in 2019, and 6.79% in 2020. The total medical expenses grew at an average annualized rate of 6.13% through 2017 and then began to level off in 2018. It increased significantly to 27% in 2018, 49% in 2019, and 19% in 2020 (Fig 3).

We examined the Gap according to the type of hearing loss (S1–S3 Files) and found that the number of people with a Gap in conductive hearing loss increased from 5.9 per 100,000 in 2011 to 9.5 per 100,000 in 2020, representing a compound annual growth rate of 5.4%. Sensorineural hearing loss increased at a CAGR of 7.3%, from 70.2 per 100,000 in 2011 to 132.6 in

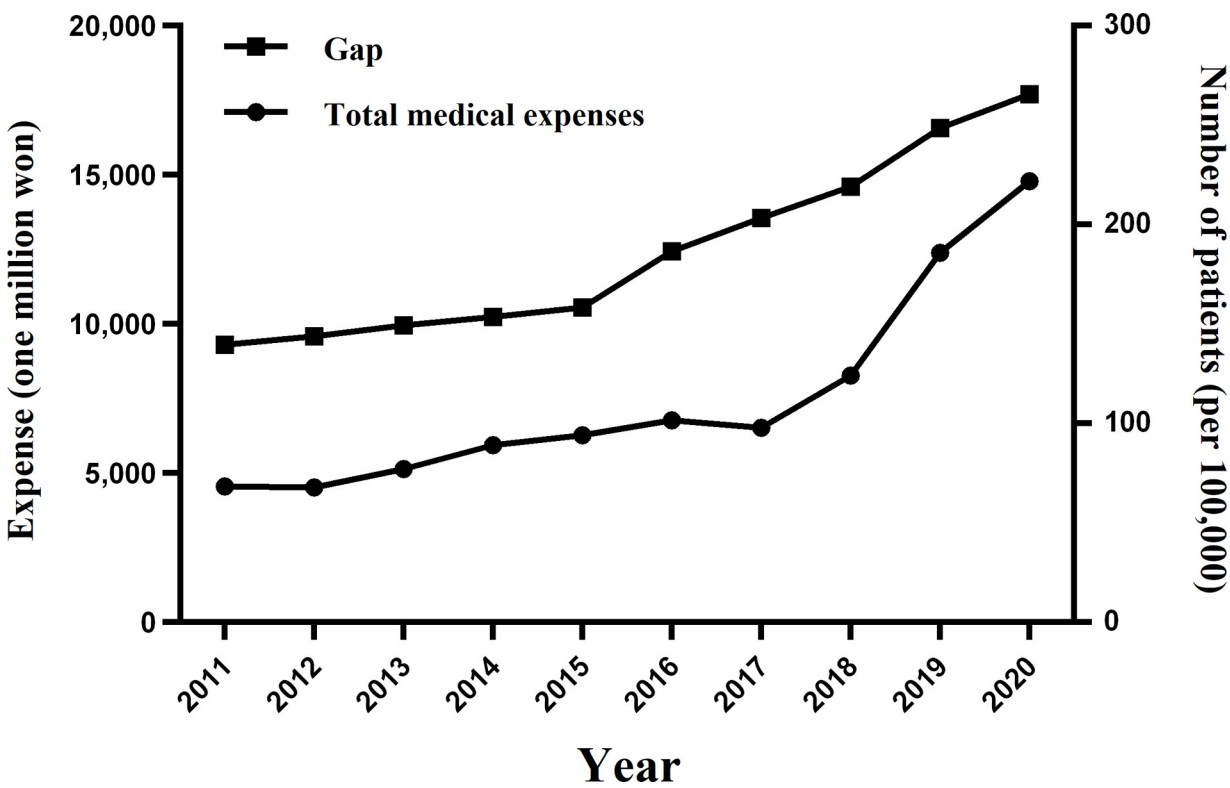

**Fig 3.**

2020. Mixed hearing loss increased at a compound annual growth rate of 8.6%, from 5.7 per 100,000 people to 12 in 2020. Ototoxicity hearing loss increased at a CAGR of -16.3%, from 0.12 in 2011 to 0.02 in 2020. Presbycusis hearing loss increased at a CAGR of 7.7%, from 3.4 in 2011 to 6.7 in 2020. Sudden hearing loss increased at a CAGR of 10.4%, from 10.3 in 2011 to 25.1 in 2020. Noise-induced hearing loss increased at a CAGR of 5.7%, from 0.7% in 2011 to 1.1 in 2020. Finally, other hearing loss increased at a CAGR of 9.5%, from 22.6 in 2011 to 51.4 in 2020. By types of hearing loss, sudden onset hearing loss had the largest average annual increase, while ototoxicity hearing loss was the only type to decrease.

Multiple regression analysis showed that the regression model showed a good fit F = 7198.92 (p = .000). Adj. R2 = 0.9988, indicating an explanatory power of 99.8% (Table 4). Sensorineural hearing loss, sudden hearing loss, and other hearing loss were statistically significant (P < .0001), while conductive hearing loss and mixed hearing loss were not significant at

**Table 4. Multiple regression by Gaps in the types of hearing loss.**

| Variable | Beta | SE | T | Pr > |t| |
|---|---|---|---|---|
| Conductive | -0.34382 | 0.44067 | -0.78 | 0.4402 |
| Sensorineural | 1.46043 | 0.07666 | 19.05 | < .0001* |
| Mixed | 0.1404 | 0.3226 | 0.44 | 0.6659 |
| Sudden | 1.01934 | 0.18913 | 5.39 | < .0001* |
| Other | 0.09923 | 0.02197 | 4.52 | < .0001* |
| $R^2$ / Adj. $R^2$ | 0.999 / 0.9988 | | | |
| F | 7198.92*(p = .000) | | | |

0.4402 and 0.6659, respectively, which were significantly higher than the significance level of 0.05. In other words, the most influential type of Gap was sensorineural hearing loss, followed by sudden hearing loss, and other types of hearing loss. The ARIMA model analysis showed that only sudden was statistically significant ($< .0001$) among the Gap types, whereas the total number of patients and the remaining types were not significant.

## Discussion

This study utilized data from the National Health Insurance Service to determine the prevalence and incidence of patients diagnosed with hearing loss in an 11-year period between 2010 and 2020. When comparing prevalence and incidence, we found that the difference between them increased annually. The authors likened this trend to a patient who never recovered from a hearing loss and whose results could be saved using unknown medical big data. This difference between prevalence and incidence is defined in this study as the "Gap." The Gap between prevalence and incidence represents the number of people with hearing loss at a given point in time who were not newly diagnosed with the condition during a specified time period. This could be because of several reasons, including people who have had hearing loss for some time but have not yet been diagnosed, people who have been diagnosed but have not received treatment or are not effectively managing their hearing loss, or people who have experienced hearing loss but have recovered to some extent and are no longer counted as new cases. Understanding the Gap between prevalence and incidence can help healthcare providers and researchers to better understand the natural history of health conditions and identify areas where improvements in diagnosis, treatment, or prevention may be required [15, 16].

According to a World Health Organization (WHO) report, more than 5 percent of the world's population 430 million people (420 million adults and 34 million children) have hearing loss and report a need for rehabilitation. It is estimated that by 2050, more than 700 million people, or one in ten, will experience hearing loss [17]. Similar to this study, a report analyzed by the Korea Health Insurance Review and Assessment Service in 2019 found that the number of people treated for hearing loss increased from 366,000 in 2009 to 583,000 in 2018, with an average annual increase of 5.3 percent [18]. Compared with this study, the increase was similar, with an average annual growth rate of 4.98% from 604,000 in 2010 to 892,000 in 2018. As of 2020, the domestic prevalence obtained in this study was 1.84%, increasing annually, and the prevalence increased with age to 4.10% among those over 60. However, the 11-year compound annual growth rate showed a large increase for people in their 20s and 30s. As of 2020, the domestic incidence was 1.57%, increasing annually, and the incidence increased with age to 3.36% for those over 60s. Similar to the prevalence rate, the 10-year compound annual growth rate showed a large increase in the 20s and 30s age group. These results confirm that the rate of increase in noise-induced hearing loss among young people had a great influence. A meta-analysis of hearing loss prevalence in Europe, unlike in South Korea where systematic epidemiologic data are available, is not well defined, in part due to the use of different classification systems. This indicates a higher prevalence than in the South Korea and raises the need for standardized procedures to collect and report epidemiological data [19].

To identify this Gap, it is important to clarify the operational definitions of prevalence and incidence. In this study, prevalence was defined as the number of patients per year based on the date of the first diagnosis, and the denominator was the total population for that year. The incidence was calculated as the number of patients with a first diagnosis of hearing loss and those with a recorded diagnosis more than 365 days after the first diagnosis of hearing loss, with the number of patients as the numerator and the total population in that year as the denominator, which is the prevalence. Therefore, the Gap represents patients who have already

been diagnosed with hearing loss and are being diagnosed annually, indicating that the number of patients who are not recovering is increasing.

To characterize the patients in the Gap, we obtained the number of patients, sex, age, and total medical expenses for care from 2011 to 2020. As of 2020, the Gap was 0.27%, showing a steady increase from 2011 to 2020 with a corresponding increase in the total medical expenses. As with the prevalence and incidence rates, there were more women than men, and a higher number of people were in their 60s. The characteristics of Gap influence the trends and patterns of prevalence and incidence. The number of patients per 100,000 by type of Gap was calculated for each sex and age group, and all types, except ototoxicity hearing loss, showed an increasing trend over the 10-year period. The type of Gap with the highest 10-year compound annual growth rate was sudden hearing loss. We organized them into eight types of prevalence and incidence as well as Gaps.

Of the types of hearing loss in the Gap, sensorineural hearing loss most affected the total number of patients. Sensorineural hearing loss was followed by sudden hearing loss and other hearing loss. We found that the number of patients diagnosed with sensorineural hearing loss who continued to visit the clinic year after year increased significantly and steadily. Studies have shown that only 8% of cases treated for sensorineural hearing loss improved [20]. In this study, I think it is the result of a well-reflected study that sensory neurotic hearing loss accounts for the most types of Gap hearing loss. A study examining the circadian rhythm (seasonal incidence) of sudden hearing loss found that it was seasonal and increased during the spring months of March, April, and May [21]. When analyzed by differencing, out of the eight types, only sudden hearing loss showed significant seasonality, with a consistent increase in the number of patients in winter. We found that patients with sudden hearing loss tended to return in the winter. Sensorineural hearing loss has become a growing social concern in aging societies. Most cases of hearing loss are incurable and permanent and require auditory rehabilitation with hearing aids [22]. Studies have shown that bone-anchored hearing aids (BAHAs) provide relief from hearing impairment and are effective in patients with single-sided deafness (SSD) [23]. Thus, Gap could be viewed as a patient constantly visiting the hospital.

To identify this Gap, it is important to clarify the operational definitions of prevalence and incidence. The Gap represents patients who have already been diagnosed with hearing loss and are being diagnosed annually, indicating that the number of patients who are not recovering is increasing. In other words, the increase in Gap meant that there were many patients who constantly visited the hospital for diagnosis of hearing loss. Studies have shown that untreated hearing loss is highly associated with higher medical costs and utilization (hospitalization and readmission) [14]. Therefore, to fully understand the burden of hearing loss and develop effective prevention and treatment strategies, it is important to measure the Gap between its prevalence and incidence. We can suppose that hearing loss is well managed by healthcare providers or is difficult to treat. Efforts to reduce the Gap between the prevalence and incidence of hearing loss involved several strategies. We can attempt to improve awareness and education regarding the importance of early detection and management of hearing loss. This can help reduce the stigma associated with seeking treatment and encourage people to seek treatment earlier, which may reduce the difference between the prevalence and incidence. We also attempted to increase access to hearing healthcare services by improving insurance coverage, increasing the number of hearing healthcare professionals, and promoting the use of telehealth services for hearing evaluation and treatment. Encouraging the early detection and treatment of hearing loss through regular hearing screenings, ensuring access to hearing aids and other assistive devices, and providing counseling and support services to people with hearing loss and their families can also help reduce the difference between prevalence and incidence. By implementing these strategies, the overall hearing health of the population can be improved

and the burden of hearing loss can be reduced. In order to conduct future research on Gap, it is necessary to explore longitudinal data over time and operational definitions of specific diseases. Not many countries have national data at the national level. For Gap research, it is necessary to collect and organize data nationwide.

## Limitation

This study has several limitations. First, we used an operationalized definition of sudden hearing loss and found that more than two-thirds of patients with sudden-onset hearing loss recover spontaneously, and the Korean healthcare system prescribes faster and more frequently than in other countries [14]. Sudden hearing loss is reversible; however, many patients develop hearing loss. In other words, patients with sudden hearing loss had a high rate of recovery; however, there was a large Gap due to patients who recovered but returned for other hearing losses or were treated periodically were treated incorrectly. However, there are limitations to this study because we did not determine the exact cause. Second, it only included patients who had been tested in person at a hospital and were diagnosed with hearing loss by a doctor; therefore, it does not include diagnoses based on a doctor's subjective judgment and people who suffer from hearing loss but do not visit a hospital [24]. Therefore, the prevalence and incidence may be low compared with studies that extrapolate from traditional sample populations. However, we believe that these data are reliable for 97% of the Korean population. Third, actual research is still limited because NHIS is used for billing purposes. Some medical staff sometimes overestimate the severity of the disease because the claims data are related to the insurance coverage item. Conversely, if it is not related to the insurance coverage item, it may be difficult to identify accurate medical coding even for serious diseases [13]. Fourth, the operational definitions of denominators and molecules for calculating prevalence and incidence as demographic limitations influence epidemiological measurements and thereby the comparability of studies. In studies conducted with various operational definitions of molecules and denominators used to calculate prevalence and incidence, the use of different denominators resulted in slight differences in incidence, and the determination of the prevalence type had a large influence on the prevalence rate [3].

## Conclusion

In conclusion, we utilized data from the National Health Insurance Service data to obtain prevalence and incidence, and defined a new group called "Gap". This Gap has steadily increased over the past decade, with sensorineural hearing loss being the most impactful. Because hearing loss is an irreversible disease, this Gap could mean many things, but it could be seen as a patient who has never recovered from hearing loss. If Gap continues to grow steadily, the economic and social burdens, including healthcare costs, will all go to patients. We propose this method to view patients who have not recovered from medical big data without test results. By using big data, we can eliminate the need for time-consuming clinical trials to save patients who have not recovered from certain diseases. This Gap can be used as a trigger for investment and interest in specific diseases. We hope that the prevalence and incidence of hearing loss obtained in this study will serve as a reference for future research on hearing loss in Korea.

## Supporting information

**S1 File. Prevalence by type of hearing loss (Number of patients per 100,000).**
(DOCX)

**S2 File. Incidence by type of hearing loss (Number of patients per 100,000).**
(DOCX)

**S3 File. Gap by type of hearing loss (Number of patients per 100,000).**
(DOCX)

## Author Contributions

**Conceptualization:** Junhun Lee, Young Joon Seo.

**Data curation:** Junhun Lee, Chul Young Yoon, Juhyung Lee.

**Formal analysis:** Junhun Lee, Chul Young Yoon, Juhyung Lee.

**Methodology:** Junhun Lee, Chul Young Yoon, Tae Hoon Kong.

**Project administration:** Junhun Lee, Tae Hoon Kong, Young Joon Seo.

**Validation:** Tae Hoon Kong.

**Visualization:** Junhun Lee.

**Writing – original draft:** Junhun Lee.

**Writing – review & editing:** Junhun Lee, Chul Young Yoon, Tae Hoon Kong, Young Joon Seo.

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
