## [Decision Letter · Decision Letter 0]

4 Jan 2024

PONE-D-23-40189Clinical characteristics of the “Gap” between the prevalence and incidence of hearing loss using National Health Insurance Service dataPLOS ONE

Dear Dr. Seo,

Thank you for submitting your manuscript to PLOS ONE. After careful consideration, we feel that it has merit but does not fully meet PLOS ONE’s publication criteria as it currently stands. Therefore, we invite you to submit a revised version of the manuscript that addresses the points raised during the review process.

We look forward to receiving your revised manuscript.

Kind regards,

Ateya Megahed Ibrahim El-eglany

Academic Editor

PLOS ONE

Journal Requirements:

2. In the online submission form, you indicated that the data analyzed in this study was obtained from the Korean National Health Insurance Service (NHIS) the following licenses/restrictions apply: only Korean researchers can access these datasets. Requests to access these datasets should be directed to NHIS, https://nhiss.nhis.or.kr/bd/ab/bdaba000eng.do.

Additional Editor Comments:

Dear Author

I trust this message finds you well. I have thoroughly reviewed your manuscript titled "Clinical Characteristics of the 'Gap' between the Prevalence and Incidence of Hearing Loss Using National Health Insurance Service Data," and I appreciate the effort and dedication you have invested in this research. The study addresses a significant gap in our understanding of hearing loss epidemiology and contributes valuable insights to the field.

However, I would like to bring to your attention several important points that require attention for a successful publication. First and foremost, the methodology section needs further elaboration, particularly concerning patient inclusion criteria, details of data cleaning processes, and the rationale behind the selected statistical methods. A more comprehensive explanation of these aspects will enhance the robustness and transparency of your study.

Additionally, it is crucial to discuss potential limitations and biases inherent in the National Health Insurance Service database. Addressing issues such as underreporting or misclassification of hearing loss cases, demographic limitations, and the database's inherent constraints will provide a more nuanced understanding of the study's results.

I also encourage you to conduct a comparative analysis with other populations or subgroups within South Korea. This additional dimension will enrich the contextual understanding of your findings and contribute to the broader discussion on hearing loss epidemiology.

Furthermore, the manuscript should explicitly address research ethics and patient confidentiality, ensuring that all necessary permissions and ethical approvals are clearly documented. Additionally, a careful review for originality and adherence to publication ethics guidelines, including authorship criteria, acknowledgment of funding sources, and declaration of potential conflicts of interest, is essential.

In terms of content, please define the term "Gap" clearly in the abstract and expound on its significance, especially in relation to healthcare systems and potential interventions. Consistency in terminology, linking the Gap increase to sensorineural hearing loss, and discussing economic and societal impacts will strengthen the overall narrative.

Finally, I encourage you to provide specific directions for future research in the conclusion, guiding the field toward further exploration and development.

I appreciate your commitment to the advancement of knowledge in this area, and I am confident that addressing these points will significantly improve the manuscript's quality. I look forward to receiving your revised submission.

Should you have any questions or require clarification on any of the points raised, please feel free to reach out.

Sincerely,

Reviewers' comments:

Reviewer's Responses to Questions

**Comments to the Author**

1. Is the manuscript technically sound, and do the data support the conclusions?

Reviewer #1: Partly

Reviewer #2: Yes

2. Has the statistical analysis been performed appropriately and rigorously? 

Reviewer #1: No

Reviewer #2: Yes

3. Have the authors made all data underlying the findings in their manuscript fully available?

Reviewer #1: Yes

Reviewer #2: Yes

4. Is the manuscript presented in an intelligible fashion and written in standard English?

Reviewer #1: Yes

Reviewer #2: Yes

5. Review Comments to the Author

Reviewer #1: This study provides important insights into the epidemiology of hearing loss and highlights a significant but underexplored area in public health. The recommendations, if addressed, could greatly enhance the impact and reliability of the findings. We look forward to seeing the revised manuscript and believe that the study has the potential to make a substantial contribution to the field.

1. Clarification of Methodology:

The methodology section provides a broad overview of the data extraction process and analysis. However, more detailed explanations are necessary for the robustness of the study. Specifically, clarifying the criteria for patient inclusion, details on data cleaning, and the rationale behind the statistical methods chosen will enhance the transparency and replicability of the study.

2. Discussion of Limitations:

While the study provides valuable insights into the prevalence and incidence of hearing loss, it is crucial to discuss potential limitations and biases. This includes limitations inherent to the National Health Insurance Service database, the potential for underreporting or misclassification of hearing loss cases, and any demographic limitations. A discussion on how these limitations might impact the findings and interpretations would provide a more balanced and cautious view of the results.

3. Comparative Analysis:

The study provides an in-depth analysis of the South Korean population. To enhance the contextual understanding of these findings, a comparative analysis with other populations or different subgroups within South Korea could be beneficial. Insights into how these findings compare with international data or different demographic groups would be valuable for readers and could guide further research.

4. Future Research Directions:

While the study concludes with several recommendations for policy and healthcare strategies, it would be beneficial to propose specific directions for future research. This could include longitudinal studies to track the "Gap" over time, investigations into the effectiveness of different treatment strategies, or explorations into the socio-economic, cultural, and behavioral factors influencing the "Gap".

Additional Comments:

5. Research Ethics and Patient Confidentiality:

It's commendable that the study utilized de-identified data. However, please ensure that all aspects of research ethics and patient confidentiality are thoroughly addressed, and that appropriate permissions and ethical approvals are clearly documented in the manuscript.

6. Dual Publication and Originality:

Ensure the work is original and has not been published elsewhere in any form or language (partially or in full). The study should cite all relevant previous work on the topic and clearly indicate how this study adds to the existing literature.

7. Publication Ethics:

The manuscript should adhere to publication ethics guidelines, including authorship criteria, acknowledgment of funding sources, and declaration of any potential conflicts of interest.

8. Language and Presentation:

Consider revising the manuscript for language clarity, grammatical correctness, and overall presentation quality. Ensuring that the manuscript is clear and well-written will make your valuable findings more accessible to a wider audience.

Reviewer #2: 1.Objectives and Gap Definition:

- Clearly define the term "Gap" in the abstract. While it's mentioned that it represents the difference between prevalence and incidence, provide a concise explanation for readers who might not be familiar with the term.

2. Significance of the Gap:

- Elaborate on the significance of the increasing Gap, particularly in terms of its implications for healthcare systems and potential interventions. This could be briefly highlighted in the abstract and expanded upon in the conclusion.

3. Consistency in Terminology:

- Ensure consistency in the use of terminology throughout the abstract and the rest of the manuscript. For instance, use the term "Gap" consistently and avoid variations that might create confusion.

4. Linking Gap Increase to Sensorineural Hearing Loss:

- Further explore and explain the connection between the increasing Gap and sensorineural hearing loss. Provide insights into why this specific type of hearing loss contributes more significantly to the observed trend.

5. Economic and Societal Impacts:

- Consider discussing potential economic and societal impacts resulting from the increasing Gap. This could enhance the broader context of the study and its relevance to policymakers and public health officials.

6. Abstract Reflecting Uniqueness:

- Ensure that the abstract clearly communicates the unique aspects of the study, such as the introduction of the term "Gap" and the emphasis on patients not recovering from hearing loss. This uniqueness should be evident to readers from the abstract itself.

7. Broadening Implications in Conclusion:

- In the conclusion, expand on the broader implications of the study's findings. Connect the increasing Gap to possible strategies for prevention, diagnosis, and treatment, emphasizing the practical applications of the research.

8. Consistency Check and Data Repetition:

• Review and clarify if the repetition of data regarding the increase in the number of people with hearing loss from 556,879 in 2011 to 806,397 in 2020 is intentional or if it requires correction.

9. CAGR Explanation:

a. Provide a brief explanation or context for the Compound Annual Growth Rate (CAGR) to assist readers in understanding the growth rates presented in the results.

10. Discussion of Age-Specific Trends:

a. Discuss any observed trends or patterns in age-specific prevalence and incidence rates, providing insights into potential implications.

11. Gender Disparities Discussion:

a. Briefly discuss any potential reasons or contributing factors for the observed differences in prevalence and incidence rates between genders.

12. Enhanced Gap Analysis:

a. Elaborate on the factors contributing to the observed gap between prevalence and incidence, discussing potential implications for healthcare systems or patient outcomes.

13. Integration of Figures and Tables:

a. Integrate key findings from tables and figures into the text to improve the flow of information and aid in the interpretation of results.

6. PLOS authors have the option to publish the peer review history of their article (what does this mean?). If published, this will include your full peer review and any attached files.

Reviewer #1: **Yes: **Mostafa shaban

Reviewer #2: **Yes: **Ateya Megahed Ibrahim

---

## [Author Response · Author response to Decision Letter 0]

5 Feb 2024

Dear Editor:

Thank you for your favorable evaluation of our research. We carefully checked what you said and revised it. The methodology section has modified or added all the overall contents. I'm really sorry, but comparative analysis with other population groups is the stage of applying for new data for the next study. We will come back to PLOS ONE with meaningful results on hearing loss dynamics through further research.

Thank you for your hard work, and I am forwarding the revised submission.

Sincerely,

Response to Reviewer 1 Comments

(Thank you for reviewing our previous version. It helped a lot.)

1. Clarification of Methodology: 

He methodology section provides a broad overview of the data extraction process and analysis. However, more detailed explanations are necessary for the robustness of the study. Specifically, clarifying the criteria for patient inclusion, details on data cleaning, and the rationale behind the statistical methods chosen will enhance the transparency and replicability of the study.

We reconstructed it to communicate more clearly. We added patient inclusion criteria, details about organizing data, and evidence for selected statistical methods.

Details on patient inclusion criteria and data cleansing: “Excluding non-reimbursable treatment for 11 years in Korea, the total number of patients extracted by diagnosis of hearing loss was 6,424,491.”, “The diagnosis name according to the classified type of hearing loss was included in the study subjects if they were treated at least once from 2010 to 2020. Therefore, there are patients who have received multiple types of hearing loss diagnoses per patient included in the study.”, “Finally, we analyzed the total number of patients in Gap and the number of patients by type”, “In order to obtain Gap, it is necessary to clarify the operational definition of prevalence and incidence. The numerator of prevalence included patients diagnosed with hearing loss in the year. The numerator of incidence was included in the corresponding year of the date of re-diagnosis of hearing loss after 365 days of the interval between diagnostic records after the first diagnosis or diagnosis. Both denominators were calculated as the total population of the Republic of Korea for the year.”

The rationale behind the statistical methods: “Multiple regression analysis was used to determine which type of patient affected the total number of Gap patients. We excluded ototoxicity, presbycusis, and noise-induced hearing loss with a small number of patients. Additionally, we used the ARIMA model, a time series analysis model, to determine whether seasonality exists in the number of patients by type of Gap. We divided each year into four seasons: March, April, and May into Spring; June, July, and August into Summer; September, October, and November into Fall; and December, January, and February into Winter since 2010. Two-sided analysis was performed, and a P value of less than 0.05 was considered to indicate significance.”

2. Discussion of Limitations: 

While the study provides valuable insights into the prevalence and incidence of hearing loss, it is crucial to discuss potential limitations and biases. This includes limitations inherent to the National Health Insurance Service database, the potential for underreporting or misclassification of hearing loss cases, and any demographic limitations. A discussion on how these limitations might impact the findings and interpretations would provide a more balanced and cautious view of the results.

Added to the " Limitation " section.

“Third, actual research is still limited because NHIS is used for billing purposes. Some medical staff sometimes overestimate the severity of the disease because the claims data are related to the insurance coverage item. Conversely, if it is not related to the insurance coverage item, it may be difficult to identify accurate medical coding even for serious diseases [14]. Fourth, the operational definitions of denominators and molecules for calculating prevalence and incidence as demographic limitations influence epidemiological measurements and thereby the comparability of studies. In studies conducted with various operational definitions of molecules and denominators used to calculate prevalence and incidence, the use of different denominators resulted in slight differences in incidence, and the determination of the prevalence type had a large influence on the prevalence rate [3].”

3. Comparative Analysis: 

The study provides an in-depth analysis of the South Korean population. To enhance the contextual understanding of these findings, a comparative analysis with other populations or different subgroups within South Korea could be beneficial. Insights into how these findings compare with international data or different demographic groups would be valuable for readers and could guide further research.

Thank you for your good opinion. The data used in this study is customized data for the diagnosis of hearing loss, and I tried for the first time with this operational definition. For more in-depth analysis, we have applied for new customized data as the next task.

4. Future Research Directions: 

While the study concludes with several recommendations for policy and healthcare strategies, it would be beneficial to propose specific directions for future research. This could include longitudinal studies to track the "Gap" over time, investigations into the effectiveness of different treatment strategies, or explorations into the socio-economic, cultural, and behavioral factors influencing the "Gap".

Added to the "Discussion" section.

“In order to conduct future research on Gap, it is necessary to explore longitudinal data over time and operational definitions of specific diseases. Not many countries have national data at the national level. For Gap research, it is necessary to collect and organize data nationwide.”

Additional Comments:

5. Research Ethics and Patient Confidentiality: 

It's commendable that the study utilized de-identified data. However, please ensure that all aspects of research ethics and patient confidentiality are thoroughly addressed, and that appropriate permissions and ethical approvals are clearly documented in the manuscript.

I've made sure it's clearly documented in this manuscript.

6. Dual Publication and Originality: 

Ensure the work is original and has not been published elsewhere in any form or language (partially or in full). The study should cite all relevant previous work on the topic and clearly indicate how this study adds to the existing literature.

I've made sure it's clearly documented in this manuscript.

7. Publication Ethics:3. Comparative Analysis: 

The manuscript should adhere to publication ethics guidelines, including authorship criteria, acknowledgment of funding sources, and declaration of any potential conflicts of interest.

I have complied with the publication ethics guidelines.

8. Language and Presentation: 

Consider revising the manuscript for language clarity, grammatical correctness, and overall presentation quality. Ensuring that the manuscript is clear and well-written will make your valuable findings more accessible to a wider audience.

Thank you for your valuable comments, we have conducted additional reviews for clear manuscripts.

 

Response to Reviewer 2 Comments

(Thank you for reviewing our previous version. It helped a lot.)

1.Objectives and Gap Definition: 

Clearly define the term "Gap" in the abstract. While it's mentioned that it represents the difference between prevalence and incidence, provide a concise explanation for readers who might not be familiar with the term.

Clarified the term "Gap" in the abstract.

“The difference between the prevalence and the incidence was defined in this study as the term "Gap". Gap is the number of patients converted into the number of patients per 100,000 people by subtracting the incidence from the prevalence.”

2. Significance of the Gap: 

Elaborate on the significance of the increasing Gap, particularly in terms of its implications for healthcare systems and potential interventions. This could be briefly highlighted in the abstract and expanded upon in the conclusion.

Overall, the "Abstract" section has been modified. 

Added content to the "Discussion" section along with references.

“In other words, the increase in Gap meant that there were many patients who constantly visited the hospital for diagnosis of hearing loss. Studies have shown that untreated hearing loss is highly associated with higher medical costs and utilization (hospitalization and readmission) [24].”

3. Consistency in Terminology: 

Ensure consistency in the use of terminology throughout the abstract and the rest of the manuscript. For instance, use the term "Gap" consistently and avoid variations that might create confusion.

Modified to be consistent with "Gap" throughout the manuscript.

4. Linking Gap Increase to Sensorineural Hearing Loss: 

Further explore and explain the connection between the increasing Gap and sensorineural hearing loss. Provide insights into why this specific type of hearing loss contributes more significantly to the observed trend.

Added content to the "Discussion" section along with references.

Studies have shown that only 8% of cases treated for sensorineural hearing loss improved [20]. In this study, I think it is the result of a well-reflected study that sensory neurotic hearing loss accounts for the most types of Gap hearing loss.

5. Economic and Societal Impacts:

Added to the "Conclusion" section.

“If Gap continues to grow steadily, the economic and social burdens, including healthcare costs, will all go to patients.”

6. Abstract Reflecting Uniqueness:

Added content to the "Abstract " section.

“In other words, the increase in Gap meant that there were many patients who constantly visited the hospital for diagnosis of hearing loss.”

7. Broadening Implications in Conclusion:

Consider discussing potential economic and societal impacts resulting from the increasing Gap. This could enhance the broader context of the study and its relevance to policymakers and public health officials.

Added to the "Discussion" section.

“In order to conduct future research on Gap, it is necessary to explore longitudinal data over time and operational definitions of specific diseases. Not many countries have national data at the national level. For Gap research, it is necessary to collect and organize data nationwide.”

8. Consistency Check and Data Repetition:

Review and clarify if the repetition of data regarding the increase in the number of people with hearing loss from 556,879 in 2011 to 806,397 in 2020 is intentional or if it requires correction.

Thank you for the correct point. I reviewed it and there was a data iteration. I modified it.

9. CAGR Explanation:

Provide a brief explanation or context for the Compound Annual Growth Rate (CAGR) to assist readers in understanding the growth rates presented in the results.

Added to the "Materials and Methods" section.

“The average annual growth rate (CAGR) was calculated by averaging the growth rate by year during the analysis period.”

10. Discussion of Age-Specific Trends: 

Discuss any observed trends or patterns in age-specific prevalence and incidence rates, providing insights into potential implications.

Added to the "Discussion" section.

“These results confirm that the rate of increase in noise-induced hearing loss among young people had a great influence.”

11. Gender Disparities Discussion:

Briefly discuss any potential reasons or contributing factors for the observed differences in prevalence and incidence rates between genders.

Added to the "Discussion" section.

“The characteristics of Gap influence the trends and patterns of prevalence and incidence.”

12. Enhanced Gap Analysis:

Elaborate on the factors contributing to the observed gap between prevalence and incidence, discussing potential implications for healthcare systems or patient outcomes.

Added content to the "Discussion" section along with references.

“In other words, the increase in Gap meant that there were many patients who constantly visited the hospital for diagnosis of hearing loss. Studies have shown that untreated hearing loss is highly associated with higher medical costs and utilization (hospitalization and readmission).”

13. Integration of Figures and Tables:

Integrate key findings from tables and figures into the text to improve the flow of information and aid in the interpretation of results.

Supplement content has been added to the "Results" section for sufficient delivery of our study.

“We examined the Gap according to the type of hearing loss (Supplement) and found that the number of people with a Gap in conductive hearing loss increased from 5.9 per 100,000 in 2011 to 9.5 per 100,000 in 2020, representing a compound annual growth rate of 5.4%. Sensorineural hearing loss increased at a CAGR of 7.3%, from 70.2 per 100,000 in 2011 to 132.6 in 2020. Mixed hearing loss increased at a compound annual growth rate of 8.6%, from 5.7 per 100,000 people to 12 in 2020. Ototoxicity hearing loss increased at a CAGR of -16.3%, from 0.12 in 2011 to 0.02 in 2020. Presbycusis hearing loss increased at a CAGR of 7.7%, from 3.4 in 2011 to 6.7 in 2020. Sudden hearing loss increased at a CAGR of 10.4%, from 10.3 in 2011 to 25.1 in 2020. Noise-induced hearing loss increased at a CAGR of 5.7%, from 0.7% in 2011 to 1.1 in 2020. Finally, other hearing loss increased at a CAGR of 9.5%, from 22.6 in 2011 to 51.4 in 2020. By types of hearing loss, sudden onset hearing loss had the largest average annual increase, while ototoxicity hearing loss was the only type to decrease.”

---

## [Editor Report · Decision Letter 1]

12 Feb 2024

Clinical characteristics of the “Gap” between the prevalence and incidence of hearing loss using National Health Insurance Service data

PONE-D-23-40189R1

Dear Dr. Joon Seo 

We’re pleased to inform you that your manuscript has been judged scientifically suitable for publication and will be formally accepted for publication once it meets all outstanding technical requirements.

Kind regards,

Ateya Megahed Ibrahim El-eglany

Academic Editor

PLOS ONE
---

## [Editor Report · Acceptance letter]

28 Feb 2024

PONE-D-23-40189R1 

PLOS ONE

Dear Dr. Seo, 

I'm pleased to inform you that your manuscript has been deemed suitable for publication in PLOS ONE. Congratulations! Your manuscript is now being handed over to our production team.

Kind regards, 

on behalf of

Dr. Ateya Megahed Ibrahim El-eglany 

Academic Editor

PLOS ONE